# Silver Nanoparticle-Induced Nephrotoxicity in Zebrafish (*Danio rerio*)

**DOI:** 10.3390/ijms26094216

**Published:** 2025-04-29

**Authors:** Grace Emily Okuthe, Busiswa Siguba

**Affiliations:** Department of Biological & Environmental Sciences, Walter Sisulu University, P/B X1, Mthatha 5117, South Africa; busiswasiguba@gmail.com

**Keywords:** silver nanoparticles (AgNPs), nephrotoxicity, Zebrafish, biosynthesis, histopathology, renal system

## Abstract

The escalating challenge of antibacterial resistance has driven the widespread use of silver nanoparticles (AgNPs) due to their potent antimicrobial properties. AgNPs can be synthesised through diverse methods, spanning conventional chemical and physical routes to the increasingly favoured biosynthesis approach. While offering environmental advantages, the ecological impact of biogenically synthesized AgNPs, especially on aquatic ecosystems, requires thorough evaluation. The renal system, critical for maintaining physiological homeostasis via nephron-mediated waste removal, fluid regulation, and electrolyte balance, is highly vulnerable to toxicant-induced damage, which can negatively affect organismal fitness. This study aimed to assess the nephrotoxic effects of AgNPs, synthesized using entirely “green” methods, on zebrafish after 96-h exposures to three distinct concentrations alongside a control group. Acridine orange fluorescence microscopy revealed dose-dependent histopathological alterations in renal tissues. Specifically, at 0.031 μg/L and 0.250 μg/L, significant changes were observed, including glomerular shrinkage, proliferation of hematopoietic tissue, dissociation and dilation of renal tubules, and melanomacrophage aggregation. At 5.000 μg/L, prolonged exposure beyond 48 h indicated a potential for renal tissue cell renewal, suggesting a possible compensatory response. These results demonstrate the sensitivity of zebrafish kidneys to AgNPs and emphasize the imperative for comprehensive in vivo toxicity testing, irrespective of synthesis method, to accurately evaluate the potential for adverse ecological impacts and ensure the preservation of aquatic ecosystem integrity.

## 1. Introduction

Nanotechnology has ushered in an era of unprecedented material innovation, with silver nanoparticles (AgNPs) emerging as a cornerstone of this revolution. Engineered within the nanoscale realm (1–100 nm), AgNPs exhibit unique physicochemical properties, including an exceptionally high surface area-to-volume ratio, that fundamentally distinguish them from their bulk counterparts [1]. This increased surface area dramatically enhances their reactivity and interaction potential, endowing AgNPs with potent antimicrobial, antiviral, and antifungal activities [2,3,4,5,6,7,8]. Consequently, their integration into diverse industrial and consumer products has proliferated, spanning medical devices [9,10], textiles [11,12,13,14,15,16], and food packaging [17,18,19,20,21,22]. Furthermore, the catalytic, biosensing, and agricultural applications of AgNPs are actively being explored, highlighting their versatility and potential to address critical technological challenges [23,24,25,26].

However, the exponential growth in AgNP production and utilization has triggered environmental and human health concerns [27]. Aquatic ecosystems, the ultimate sink for numerous pollutants, are particularly vulnerable to AgNP contamination. Wastewater treatment plant effluents, industrial discharges, and runoff from diverse consumer product applications are primary conduits for AgNP entry into rivers, lakes, and oceans [28]. Even at low concentrations, the continuous influx of these nanomaterials raises critical questions about their long-term ecological impacts and the potential for cumulative toxicity in aquatic organisms [29].

While advantageous for technological applications, the unique physicochemical properties of AgNPs also contribute to their potential toxicity. Their nanoscale dimensions facilitate cellular uptake and intracellular interactions, potentially disrupting vital biological processes. The high surface reactivity of AgNPs can induce oxidative stress and inflammatory responses in exposed organisms [30,31]. Furthermore, releasing silver ions (Ag+) from dissolved AgNPs, influenced by environmental factors like pH and organic matter content, poses an additional threat to aquatic life [29,32]. The potential for bioaccumulation and biomagnification of AgNPs along aquatic food chains raises further concerns about long-term ecological impacts and risks to human health through seafood consumption [33,34,35].

AgNPs have been shown to induce toxicity in zebrafish embryos, causing mortality, hatching delays, and developmental abnormalities [36,37]. AgNPs can accumulate in various organs, including the brain, heart, and blood. Chronic exposure in adult zebrafish leads to histological changes in gills and gut, with increased expression of metallothioneins [38]. The toxicity of AgNPs is influenced by their size, shape, and aggregation state rather than concentration alone [38,39]. Notably, AgNPs can be more toxic than silver ions (Ag+) [39], although both forms deplete glutathione levels and induce oxidative stress [37]. The shape of AgNPs also affects toxicity, with flat nanoparticles showing more significant harm than spherical ones. These findings highlight the complex nature of AgNP toxicity in aquatic organisms [40,41].

The zebrafish (*Danio rerio*) has emerged as a premier model organism in aquatic toxicology owing to its well-characterized genetics, rapid development, optical transparency during early life stages, and conserved organ systems, including the kidney, which shares significant homology with mammalian renal structures [42,43]. Thus, the zebrafish kidney is an ideal target for investigating nephrotoxicity induced by nanomaterials [44].

Therefore, this study aims to comprehensively investigate the effects of “green”-synthesized AgNPs on zebrafish kidneys, utilizing acridine orange fluorescence microscopy to elucidate cellular responses. By addressing these questions, we aim to contribute to a more comprehensive understanding of the environmental risks posed by AgNPs, informing the development of sustainable nanotechnology practices and safeguarding the health of aquatic ecosystems. Ultimately, this research will highlight the urgent need for a paradigm shift in nanotoxicology, emphasizing the integration of advanced molecular and cellular techniques to accurately assess the long-term impacts of engineered nanomaterials on aquatic organisms and ecosystems.

## 2. Results

### 2.1. Pathological Observations in the Kidney: H&E Stain

For the duration of the study, there were no fish mortalities. Control groups displayed normal histological structure of renal tubules in kidney tissues (Figure 1A). The most common pathological changes in AgNP-treated groups were in the epithelium of renal tubules. In contrast, changes in glomeruli architecture were comparatively less severe. Changes such as glomerular shrinkage, blood congestion, increase in hemopoietic cells between tubules, elongation of kidney tubules and tubular necrosis were common. After exposure to 0.031 μg/L of AgNPs, injuries to renal tissues were mild to moderate at 24 h (Figure 1B). Still, with an increase in the infiltration of hematopoietic cells between tubules, intensely basophilic clusters of cells were noted in the interstitium with large conspicuous nuclei.

At 48 and 96 h, at the same concentration of AgNPs, severe necrosis and elongation of tubules were common; cells with hypertrophied nuclei were detected. Shrinkage of glomeruli, disorganized tubules, and hyaline degeneration of tubular epithelium, including disorganization of hematopoietic cells, were common (Figure 1C,D). At 96 h, nuclei of epithelial cells were small and less conspicuous and, in some cases, not visible in tissue sections. After exposure to 0.250 μg/L of AgNPs, changes in the histo-architecture of the renal tissues were visible, with very few hematopoietic cells in between tubules (Figure 1E). However, after 24 h, nuclei of epithelial cells were condensed and conspicuous in tissue sections (Figure 1E–G).

At 5.000 μg/L of exposure to AgNPs, early signs of tubular restoration were evident (Figure 1H–J), indicative of renal tubule recovery from AgNP-induced injury. In addition to tubular regeneration, there was a resurgence in the infiltration of hematopoietic cells between tubules. Occasionally, eosinophilic developing tubules were seen in groups, and developing glomeruli were present in kidney tissue sections of the treated fish (Figure 1H–J). Nephrotoxicity was observed in exposed fish, which increased in severity with exposure levels up to 48 h of exposure.

### 2.2. Fluorescent Staining: Acridine Orange (AO)

To estimate the level of cell death in Paraplast embedded kidney tissue sections, the intensity of AO fluorescence was measured semi-quantitatively. AO is cell-permeable and interacts with DNA and RNA by intercalation or electrostatic attractions, which allows the identification of engulfed apoptotic cells because it will fluoresce upon engulfment. Accordingly, cytoplasmic AO correlates with RNA, whereas 90% of nuclear staining results from DNA. Fish were exposed to 0, 0.031, 0.250 and 5.000 μg/L of AgNPs for 24, 48 and 96 h, as mentioned above. The results shown in Figure 2 indicated distinct differences between the exposure and control groups. Green, isolated yellow, and orange-stained nuclei were observed representing normal, early apoptotic and late apoptotic cells, respectively. The yellow staining in Figure 2c–e indicated cells at an early stage of apoptosis. The typical apoptotic changes, such as condensed chromatin and fragmented nuclei, were also observed after exposure to 0.031 μg/L AgNPs at 48 and 96 h. In Figure 2f,g additional features of sporadic apoptotic bodies of orange necrotic cells were found, indicating that renal tissues and cells were in the final stages of apoptosis after exposure to 0.250 μg/L of AgNPs at 24 to 48 h, respectively. At 96 h, aggregates of yellow clusters were seen around renal tubules (Figure 2h). At 5.000 μg/L AgNPs, the highest concentration, aggregates of orange clusters were seen around renal tubules at 24 h, with the reappearance of brilliant green hematopoietic cells. However, nuclei of glomeruli cells revealed yellow fluorescence. This was followed by a decrease in orange fluorescent nuclei at 48 h (Figure 2j). The nuclei of glomeruli exhibited green fluorescence. At 96 h, green fluorescent nuclei predominated (Figure 2k).

## 3. Discussion

Toxicity testing of AgNPs is crucial to understanding their potential adverse effects on biological systems and the environment [45]. These tests are conducted using various in vitro (cell-based) [46] and in vivo (animal-based) [47] models to assess different aspects of toxicity. Toxicity testing of AgNPs assesses their diverse adverse impacts on biological systems, including cytotoxicity (cell damage or death, evaluated through viability, membrane integrity, and metabolic activity), genotoxicity (DNA damage leading to potential mutations or chromosomal aberrations, measured by micronucleus and comet assays), induction of oxidative stress (production of reactive oxygen species causing cellular damage), and inflammatory responses. Additionally, the research examines organ-specific toxicity [48,49] by analyzing AgNP accumulation and harmful effects in organs like the liver, kidneys, lungs, brain, and reproductive systems, alongside their biodistribution and potential for bioaccumulation, which is vital for understanding long-term risks.

AgNP toxicity is significantly influenced by pH [50], which affects nanoparticle stability, aggregation, and silver ion release, consequently altering their biological interactions and overall toxicity. Smaller AgNPs, often formed at higher pH, generally exhibit greater toxicity due to increased surface area and cellular uptake, a relationship modulated by pH [50]. Shape, pH-dependent surface charge, and coatings further impact toxicity. Higher concentrations and exposure routes, such as oral intake (as in the current study), can lead to varying biodistribution and effects [51], with the gut being a primary site of impact. The gut’s pH environment during oral exposure (as highlighted in the current study) is critical, affecting AgNP stability and absorption (smaller particles tend to be absorbed more), potentially causing systemic toxicity like DNA damage and organ effects, with the kidney often showing significant nanoparticle accumulation and adverse effects [51]. Overall, oral AgNP toxicity is determined by nanoparticle properties, dose, pH, and the experimental model.

### 3.1. The Emerging Threat of Silver Nanoparticles in Aquatic Ecosystems

The increasing integration of silver nanoparticles (AgNPs) into various consumer products has become a prominent feature of modern manufacturing [52]. Their potent antimicrobial properties have driven their incorporation into medical devices, household items, and even water treatment systems. In 2015 alone, over 410 products on the global market contained AgNPs, with an estimated annual global production exceeding 550 tons, indicating the scale of their utilization and potential environmental release [52]. This widespread application inevitably leads to the introduction of AgNPs into various environmental compartments, with aquatic ecosystems as a significant sink for these engineered nanomaterials [53].

Despite the growing prevalence of AgNPs in consumer products and the consequent likelihood of their presence in aquatic environments [29], specifically in fish embryos [54,55,56], a comprehensive understanding of their toxicological interactions at the nanoscale within these ecosystems remains limited [52]. While numerous studies have explored the biological uptake, the precise mechanisms by which these nanoparticles exert their toxicity are still not fully elucidated.

The present study addresses this critical knowledge gap by evaluating the in vivo histological effects of environmentally synthesized, or ‘green’, AgNPs on the kidney tissue of zebrafish. The use of ‘green’ synthesized nanoparticles is a significant aspect of this research, as it explores the potential for environmentally friendly synthesis methods to reduce the inherent toxicity often associated with nanomaterials. The central hypothesis guiding this investigation was that even AgNPs synthesized using environmentally benign methods would induce significant nephrotoxicity in zebrafish, a well-established model organism in toxicological research due to its conserved physiology with other vertebrates and the ease with which its developmental processes can be studied [57].

### 3.2. Histopathological Alterations in Zebrafish Kidneys: Morphological Evidence of Nephrotoxicity

The initial assessment of zebrafish kidneys exposed to the ‘green’ synthesized AgNPs revealed moderate histopathological alterations within a relatively short exposure period of 48 h. This rapid onset of tissue damage suggests a swift interaction between the nanoparticles and the kidney cells, leading to structural changes. These early alterations were specifically observed in the renal tubule epithelium, the functional unit responsible for filtration and reabsorption, and in Bowman’s space, the area surrounding the glomerulus where initial blood filtration occurs. The involvement of these key nephron components indicates a broad impact of the AgNP exposure on renal tissue.

Interestingly, the study also documented an increase in tubule formation within the zebrafish kidneys at the 96-h time point following AgNP exposure. This later observation suggests a dynamic response of the kidney tissue, where the initial damage may trigger compensatory mechanisms for repair or regeneration. The increased formation of tubules could represent either the proliferation of existing tubule cells to replace damaged ones or the generation of entirely new nephron structures. The findings of this study, particularly the renal pathologies observed after AgNP exposure, align with previous reports in the scientific literature that have investigated the effects of nanoparticle exposure on the kidneys of fish and other model organisms, as stated in [58]. This consistency across different studies and with various types of nanoparticles strengthens the evidence that nanoparticle exposure, including AgNPs, can induce nephrotoxic effects in aquatic vertebrates, highlighting a potential widespread environmental concern.

Further research has provided more specific details regarding the nature of the renal tubule epithelial damage observed in zebrafish exposed to AgNPs. For instance, a study on adult zebrafish that were fed AgNPs (16.6 nm) for 56 days reported the occurrence of necrosis, which is premature cell death, and the detachment of the mucoid columnar epithelium lining the kidney tubules [59]. This indicates a direct cytotoxic effect of the AgNPs on the structural integrity of the tubules. It is also recognized that cell damage and tubular necrosis are common responses in teleost fish kidneys when exposed to various xenobiotic substances, likely due to the high capacity of these cells for membrane transport, which can lead to the accumulation of toxic compounds within the tubule lumen [59]. These specific types of damage, such as cellular death and the loss of the epithelial lining, directly compromise the ability of the renal tubules to effectively carry out their crucial functions in maintaining fluid and electrolyte balance within the organism.

### 3.3. Alterations in Bowman’s Space and Implications for Glomerular Filtration

The current study’s observation of changes in Bowman’s space within 48 h of exposure to ‘green’ synthesized AgNPs is a significant finding, as Bowman’s space is an integral component of the glomerulus, the primary filtration unit of the kidney. This fluid-filled space within Bowman’s capsule surrounds the glomerular capillaries, where blood filtration occurs. Alterations in this space can directly impact the efficiency and selectivity of this crucial process.

The widening of Bowman’s space could indicate a disruption in the structural relationship between the glomerulus and its surrounding capsule, potentially affecting the pressure dynamics and overall filtration efficiency [59]. Any alteration to these structures, such as flattening or fusion, can significantly impair the selectivity of the filtration process, potentially allowing larger molecules, like proteins, to pass through into the filtrate. The glomerular filtration barrier (GFB), which includes the glomerular capillaries, the glomerular basement membrane, and the podocytes with their pedicels, acts as a highly specialized nanoscale sieve [60]. The size of nanoparticles is critical in determining their ability to interact with this barrier. Research indicates that the GFB has a size cut-off of approximately 8–10 nm or 30–50 kDa for the passage of nanoparticles. Nanoparticles smaller than this threshold, such as those used in the current study with a nanoparticle size of 3.76 nm, can generally cross the GFB and enter the renal tubule system, where they may be cleared from the body through glomerular filtration [60]. Therefore, the size characteristics of the ‘green’ synthesized AgNPs used in the current study are crucial for understanding their potential to directly interact with the glomerulus and Bowman’s space. If the AgNPs are within the size range that allows them to pass through the GFB, they could directly access the structures within Bowman’s capsule, including the podocytes and their pedicels, potentially leading to the observed structural alterations and leading to a decrease in the glomerular filtration rate (GFR). Damage to the glomerular filter is a well-established trigger for acute kidney injury (AKI) [44]. A reduction in GFR impairs the kidney’s ability to effectively filter waste products from the blood, which can lead to their accumulation in the body and disrupt overall homeostasis. The flattening or fusion of pedicels, as reported in other studies involving AgNP exposure, could further exacerbate this issue by disrupting the precise size and charge selectivity of the filtration slits, potentially allowing larger molecules to pass into the filtrate and further compromising kidney function [59]. Thus, the AgNP-induced alterations in Bowman’s space, particularly those affecting the delicate structure of the podocytes, have significant implications for the kidney’s primary function of blood filtration.

### 3.4. Renal Tubulogenesis in Response to Nanoparticle Exposure: A Regenerative Attempt?

The observation of increased tubule formation in the zebrafish kidneys at 96 h following exposure to the ‘green’ synthesized AgNPs strongly suggests that the organism is actively attempting to repair the damage induced by the nanoparticles, as reported by Mckee and Wingert [44]. Teleost fish, including the zebrafish, are renowned for their remarkable capacity to regenerate kidney tissue, including the formation of new nephrons, a process known as de novo nephrogenesis, which can occur throughout their lifespan and is significantly enhanced in response to injury. This regenerative prowess contrasts with mammals, which cannot generate new nephrons in adulthood.

Kidney regeneration in zebrafish involves an initial phase of cell death and the detachment of damaged tubular cells from the basement membrane. This is followed by the proliferation of either surviving, dedifferentiated cells or, more significantly, resident stem or progenitor cells within the kidney tissue [61]. Zebrafish kidneys harbour a population of nephron progenitor cells located in the interstitial stroma, which can be activated in response to injury, such as that caused by AgNP exposure, to generate new nephrons [44]. These progenitor cells can migrate to the site of damage, forming clusters and then undergoing differentiation to develop into fully functional nephron segments. The robust tubular regeneration observed at 96 h in the current study is likely a result of the activation, proliferation, and subsequent differentiation of these resident renal progenitor cells within the zebrafish kidney in response to the injury caused by the ‘green’ synthesized AgNPs.

### 3.5. Inflammatory Cell Infiltration and Haematological Response

The observation of notable inflammatory cell infiltration in the zebrafish kidney tissue following exposure to the ‘green’ synthesized AgNPs indicates that these nanoparticles trigger an active immune response within the kidney. This is consistent with the understanding that metal nanoparticles can be recognized as foreign materials by biological systems, often leading to the activation of immune cells and the initiation of inflammatory processes. The recruitment of these inflammatory cells from the bloodstream to the site of injury in the kidney is a clear indication of haematological response to metal exposure [62].

Research on the effects of AgNP exposure in zebrafish has demonstrated a specific pattern of changes in the circulating immune cell populations, characterized by an increase in neutrophils and a decrease in lymphocytes [63]. Neutrophils are white blood cells that play a crucial role in the early stages of inflammation, acting as the first responders to injury or infection. On the other hand, lymphocytes are involved in the adaptive immune response, a more specific and long-lasting form of immunity. The increase in neutrophils in the kidney tissue of the AgNP-exposed zebrafish suggests an acute inflammatory response, likely aimed at clearing the nanoparticles from the tissue or initiating the repair of any damage caused.

### 3.6. Molecular Mechanisms Underlying Teleost Kidney Regeneration After Nanoparticle Damage

The remarkable ability of teleost fish, such as zebrafish, to regenerate their kidneys after injury relies on the activation of molecular pathways that are fundamentally conserved with those involved in kidney development (organogenesis) across vertebrates [61]. This conservation suggests that the basic building blocks and regulatory mechanisms for kidney formation and repair have been maintained throughout evolution. Following acute kidney injury, zebrafish exhibit a dual regenerative capacity, involving both the replacement of damaged epithelial cells in existing nephrons and the generation of entirely new nephrons, a process called de novo nephrogenesis [61]. This ability to form new nephrons in adulthood is a key difference compared to mammals, where nephrogenesis largely ceases after birth.

The cellular mechanisms of kidney regeneration in zebrafish involve an initial phase where damaged cells die and detach from the structural framework of the nephron. This is followed by the activation of resident renal progenitor cells [44]. These progenitor cells, which reside in the interstitial space between the nephrons, can migrate to the site of injury, undergo rapid proliferation to increase their numbers, and then differentiate into the specialized epithelial cells that form the various segments of the nephron. Studies have successfully identified and characterized specific populations of adult nephron progenitor cells in zebrafish that can generate new, functional nephrons when transplanted into recipient fish, demonstrating their stem-cell-like properties [64]. The intense tubular regeneration observed in the current study likely involves the activation and proliferation of these identified adult nephron progenitor cells in response to the damage caused by the ‘green’ synthesized AgNPs. Future research could focus on pinpointing the specific molecular signals that trigger the activation of these progenitor cells following AgNP exposure, potentially uncovering key regulators of kidney regeneration. Furthermore, given that AgNP exposure is known to induce oxidative stress in zebrafish [63], the interplay between this initial damage-inducing stress and the subsequent activation of kidney regeneration warrants further investigation to understand the full biological response.

### 3.7. Synergistic Effects of Silver Nanoparticles and Other Environmental Stressors

The current study astutely highlights the potential for the nephrotoxic effects of AgNP exposure to be significantly amplified or modified by other environmental stressors [53]. This is a crucial consideration because aquatic organisms in their natural habitats are rarely exposed to single pollutants in isolation. Instead, they typically encounter complex mixtures of various contaminants, including heavy metals, pesticides, pharmaceuticals, and microplastics, alongside engineered nanomaterials like AgNPs [53]. These co-occurring stressors can interact with each other in numerous ways, potentially leading to additive effects, where the combined toxicity is the sum of the individual toxicities; synergistic effects, where the combined toxicity is greater than the sum of the individual toxicities; or even antagonistic effects, where the combined toxicity is less than what would be expected from the individual components. Research has already begun to explore some of these potential synergistic interactions. For example, studies have shown that polystyrene nanoplastics can mediate the toxicity of silver nanoparticles in zebrafish embryos [65]. The nanoplastics, due to their large surface area, can adsorb AgNPs and potentially act as carriers, facilitating their uptake by the organism or altering their distribution and subsequent toxicity within the tissues. This interaction could lead to an enhanced accumulation of AgNPs in certain organs, potentially exacerbating their toxic effects on the kidney or other systems. Conversely, investigations into the combined effects of AgNPs and mercury on zebrafish have revealed more complex scenarios, with some findings suggesting a potential reduction in mercury toxicity in the presence of AgNPs, possibly due to the binding of mercury ions to the surface of the nanoparticles [66]. This indicates that the nature and extent of synergistic effects can vary greatly depending on the specific combination of pollutants involved and the biological endpoints being measured.

Temperature is another critical environmental factor that can influence the toxicity of pollutants in aquatic organisms [67]. As ectotherms, zebrafish physiology and metabolism depend highly on ambient water temperature. Temperature changes can affect various processes that influence pollutant toxicity, including the rate of uptake, distribution within the organism, metabolic transformation, and excretion of nanoparticles like AgNPs. Furthermore, temperature fluctuations can also act as a stressor on their own, potentially making the organisms more susceptible or more resilient to the effects of other pollutants. While the current study was conducted under controlled temperature conditions, it is essential to recognize that environmental temperature in aquatic habitats can vary significantly due to seasonal or climate changes. Future research should, therefore, consider the potential influence of temperature variations on the nephrotoxic effects of AgNP exposure in zebrafish to gain a more comprehensive understanding of the ecological risks under different environmental scenarios.

### 3.8. Advanced Methodologies for Differentiating Apoptosis and Necrosis in Renal Damage

While acridine orange (AO) staining, as mentioned in the study, can provide valuable insights into the general health of cells and indicate the presence of cell death, it has limitations in definitively distinguishing between the specific mechanisms of cell death, particularly apoptosis and necrosis, and in providing quantitative data. Apoptosis is a programmed cell death characterized by specific biochemical and morphological changes, often occurring in response to cellular stress or damage. Necrosis, conversely, is typically considered an unprogrammed form of cell death resulting from severe cellular injury. To gain a more detailed and mechanistic understanding of how AgNPs induce renal damage in zebrafish, advanced methodologies are required to identify and quantify the extent of both apoptosis and necrosis.

Several quantitative techniques are available for the detection and measurement of apoptosis. The TUNEL assay (Terminal Deoxynucleotidyl Transferase dUTP Nick End Labeling) is a widely used method that detects the fragmentation of DNA, a hallmark event in apoptosis, by enzymatically labeling the free ends of DNA fragments [68]. Using fluorescence microscopy, this assay can be used on tissue sections to visualize and quantify apoptotic cells. Another powerful technique is the Caspase 3 assay. Caspase 3 is a key executioner protease in the apoptotic pathway, and its activation is a reliable indicator of apoptosis [69]. Antibodies specific to the activated form of Caspase 3 can be used in immunohistochemistry or immunofluorescence to identify and count apoptotic cells in tissue samples. Annexin V staining is another valuable method that detects the translocation of phosphatidylserine from the inner to the outer leaflet of the plasma membrane, an early event in apoptosis. Fluorescently labeled Annexin V can be used with flow cytometry or fluorescence microscopy to identify apoptotic cells.

For the quantitative detection of necrosis, the LDH release assay (Lactate Dehydrogenase) is a commonly used biochemical method [70]. Necrosis is characterized by the loss of plasma membrane integrity, which releases intracellular enzymes like LDH into the surrounding medium. Measuring the activity of LDH in the culture medium or tissue homogenates provides a quantitative measure of cell death due to necrosis. Propidium iodide (PI) staining can detect necrotic cells [42]. PI is a fluorescent dye that cannot penetrate intact cell membranes. Therefore, cells that stain positive for PI are those with damaged membranes, indicative of necrosis.

### 3.9. Long-Term Ecological Implications and Future Research Directions

The observation that zebrafish kidneys exhibit susceptibility to AgNP-induced damage, even when the nanoparticles are synthesized using environmentally friendly methods, underscores a critical vulnerability of aquatic organisms to engineered nanomaterials. While the study noted a potential for recovery within 96 h under controlled laboratory conditions, the initial damage and the subsequent activation of repair mechanisms raise significant concerns about the long-term ecological implications of chronic exposure to AgNPs. The continuous release of AgNPs into aquatic environments from various consumer products and industrial processes suggests that aquatic organisms are likely to experience prolonged exposure to these nanoparticles, potentially leading to cumulative toxic effects that may not be apparent in short-term studies. The potential for these effects to be compounded by other environmental stressors further amplifies the need for a thorough understanding of the long-term consequences.

In light of these concerns, future research should prioritize elucidating the precise molecular mechanisms underlying AgNP-induced renal toxicity in zebrafish, focusing on differentiating between the roles of necrotic and apoptotic pathways. This mechanistic understanding is crucial for accurately assessing the risks posed by AgNPs and developing effective strategies to mitigate their potential harm to aquatic ecosystems. Furthermore, it is essential to move beyond short-term histological assessments and investigate the long-term ecological impacts of AgNP exposure on zebrafish, including effects on their reproductive success, developmental processes, and overall survival under more environmentally realistic conditions. To address these critical questions, future research should prioritize designing and implementing long-term, multi-generational experiments that simulate realistic environmental conditions, including chronic low-level AgNP exposure and other relevant environmental stressors such as co-occurring pollutants and temperature fluctuations.

These studies should integrate various methodologies, including advanced biochemical and molecular techniques, to elucidate the intricate mechanisms of nanoparticle-induced cellular damage and the subsequent repair processes. Investigating the potential for cumulative toxic effects resulting from chronic, low-level AgNP exposure is crucial, particularly for other environmental stressors that aquatic organisms routinely encounter. Understanding the complex interplay between nanoparticle properties (including size, shape, and surface charge), biological responses at various levels of organization, and the surrounding environmental context (such as water chemistry, temperature, and the presence of natural organic matter) is essential for developing sustainable nanotechnology practices and for safeguarding the health of aquatic ecosystems.

## 4. Material and Methods

### 4.1. Zebrafish Maintenance

Adult zebrafish of undefined commercial strain were purchased from a local dealer (Northland Pets, Durban, South Africa). Fish were acclimatized for at least two weeks in a 40 L stock aquarium, tanks supplied with dechlorinated tap water and aerators to ensure sufficient oxygen dissolution. Fish were maintained on a 14:10 h light: dark cycle at 28.4 °C. Mature fish were fed twice daily with freshwater Aquarium Flakefood (TetraMin, Tetra GmbH, Hamburg, Germany). Uneaten food was siphoned from tanks before fresh food was given.

### 4.2. Chemicals

The aqueous AgNPs used in this in vivo study on adult zebrafish were synthesized and characterized by Oluwafemi et al. [71], with a TEM-determined average diameter of 3.76 ± 1.00 nm.

### 4.3. Toxicity Testing of AgNPs

Adult mature male and female zebrafish (standard length 28.1 ± 0.2 mm) were exposed to nominal concentrations of AgNPs at 0 (control), 0.031 μg/L (low), 0.250 μg/L (medium), and 5.000 μg/L (high) for 96 h under 24 h static renewal system. The control group received aquarium water. The concentrations used in the current study were based on the previously published information on the LC50 value of AgNPs for 96 h in Japanese medaka (*Oryzias latipes*), which was 0.9 mg/L, [72]; the value for adult and larvae of zebrafish at 7.07 mg/L and 7.20 mg/L, respectively [73], and for fathead minnow (*Pimephales promelas*) were 1.36 mg/L [74]. Exposure treatments were conducted in 20 L aerated glass tanks, into which doses of AgNP solutions were introduced. Fifteen randomly selected adult fish were exposed to each concentration and control (i.e., five fish per replicate). During the exposure period, fish were not fed, given that nanoparticles would adhere to food particles. Debris was siphoned from each experimental unit daily. At 24, 48, and 96 h of exposure, 5 fish from each concentration and the control group were used as samples for histopathological analysis.

### 4.4. Sample Fixation and Tissue Sectioning

Fish from each concentration were sacrificed by anaesthetizing with tricaine methanosulfonate (MS222^®^, Sigma-Aldrich, St. Louis, MO, USA) at 4.2 mL tricaine stock solution in 100 mL tank water, as described by [75]. Fish were individually measured for total length, and the trunk and head regions were cut out before fixation. The trunk and head regions were fixed in Davidson’s solution overnight at room temperature, rinsed briefly with tap water, and then transferred directly to 70% ethanol. All fixed tissues were dehydrated through ethanol series, from 80% to 100%, and cleared in methyl salicylate (Sigma-Aldrich, St. Louis, MO, USA) before being embedded in Paraplast wax (Merck, Darmstadt, Germany). Paraplast blocks were trimmed to the tissue surface, and sections were cut at 3–5 μm thickness. Tissue sections were then floated in a water bath at 37° C and placed on glass slides pre-treated with 2% 3-amino-propyl triethoxysaline (Sigma-Aldrich, St. Louis, MO, USA) in acetone. Tissue sections were then dewaxed in xylene, hydrated in ethanol series [75], stained with Mayer’s modified haematoxylin, and counterstained with eosin.

To investigate the role of apoptosis/necrosis in AgNP toxicity in renal tissues, consecutive sections of the same tissues were processed and stained with acridine orange (AO) [42]. Briefly, tissue sections were hydrated and then stained in AO (Sigma Chemicals, St Louis, MO, USA) made up of distilled water and acetic acid for 30 min, rinsed in 0.5% acetic acid in 100% ethanol, and finally rinsed in two additional changes of 100% ethanol. Sections were then rinsed in two changes of HistoChoice^®^ Clearing (Merck, Darmstadt, Germany) agent and mounted on a glass slide using a histomount mounting medium. Acridine orange is a nucleic acid selective metachromatic dye, which emits green fluorescence upon intercalation with DNA and is widely used for detecting the sites of apoptosis in zebrafish. Acridine orange can permeate apoptotic cells and bind to DNA. In contrast, normal cells are non-permeable to acridine orange images of stained material were acquired using a Leica DM 750 fluorescent microscope (Leica Microsystems, GmbH, Hamburg, Germany), attached to a DFX 310 FX digital camera, and analyzed using Leica LAS imaging software version 4.5.

### 4.5. Ethical Approval

The Animal Ethics Screening Committee Faculty of Natural Sciences, Walter Sisulu University, approved the experimental protocols for the study.

## 5. Conclusions

This study provides evidence demonstrating the susceptibility of zebrafish kidneys to damage induced by even ‘green’ synthesized silver nanoparticles, underscoring that environmentally benign synthesis methods do not guarantee the absence of potential biological harm at the nanoscale. While the observed recovery potential within 96 h under controlled laboratory conditions suggests a degree of resilience, the initial histopathological alterations and the subsequent activation of regenerative processes highlight a clear biological impact of AgNP exposure on this vital organ system. The dose-dependent nature of these changes further emphasizes the need for careful consideration of the levels of AgNP contamination in aquatic environments.

The seemingly transient nature of the renal tissue responses observed in this study should not diminish the concern for the long-term ecological implications of chronic AgNP exposure. As demonstrated here, the critical vulnerability of aquatic organisms to engineered nanomaterials raises significant questions about the potential for sublethal effects to accumulate over time, especially when organisms are simultaneously exposed to other environmental stressors. These cumulative effects could have far-reaching consequences for individual organism health, population dynamics, and the overall stability of aquatic ecosystems.

The findings of this study underscore the necessity for a paradigm shift in the field of nanotoxicology. Future investigations must move beyond simple assessments of mortality and integrate advanced biochemistry and molecular biology methodologies to fully elucidate the intricate mechanisms by which nanoparticles interact with biological systems at the cellular and molecular levels. A deeper understanding of these mechanisms, including the precise pathways of cellular damage and repair, is crucial for developing accurate risk assessments and effective mitigation strategies.

In the face of increasing nanomaterial contamination, safeguarding planetary health requires a concerted effort focused on future research that prioritizes long-term, multi-generational studies under environmentally realistic conditions. Understanding the complex interplay between nanoparticle properties, biological responses, and the multifaceted environmental context is essential for developing sustainable nanotechnology practices and ensuring the long-term health and biodiversity of our planet’s vital aquatic resources.

## Figures and Tables

**Figure 1 ijms-26-04216-f001:**
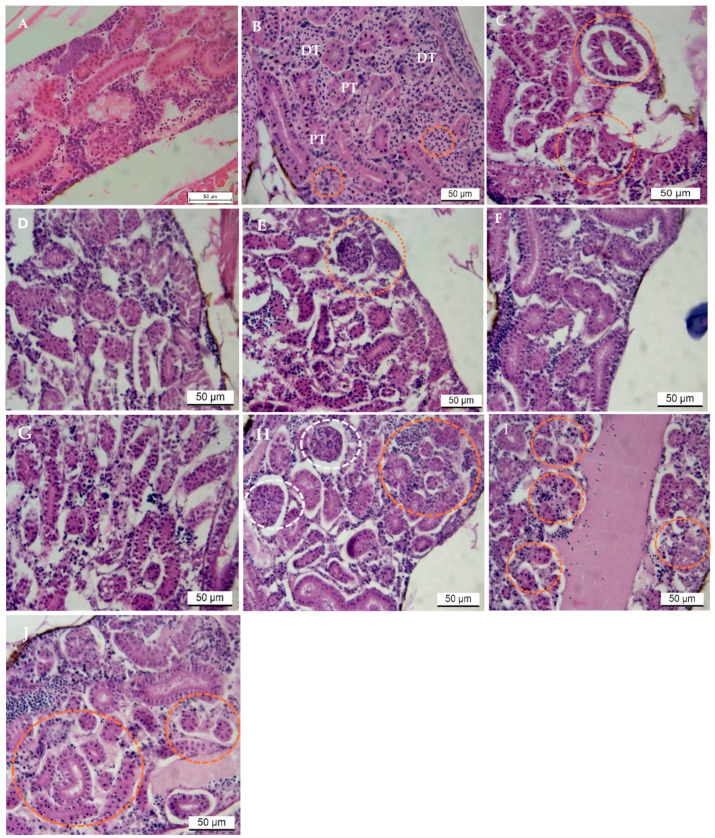
Haematoxylin and eosin (H&E) stained sections of zebrafish kidney after exposure to various concentrations of silver nanoparticles (AgNPs) for 24 to 96 h. (**A**) Control section with normal renal tissue architecture. (**B**–**D**) shows the tissue section of fish exposed to AgNPs at 0.031 μg/L at 24, 48 and 96 h, respectively. (**B**) shows infiltration of cell at 24 h (orange dotted circles), tubules (proximal (PT) and distal (DT), and intensely basophilic cell clusters in the interstitium. (**C**,**D**) shows disorganized kidney tubules (orange dotted circles) and loss of tissue architecture and the overall distortion of the histo-architecture of renal tubules. (**E**–**H**) shows tissue sections of fish exposed to AgNPs at 0.250 μg/L of AgNPs at 24, 48 and 96 h, respectively, orange circles in E, indicate shrunken glomeruli. (**H**–**J**) shows sections of tissues exposed to 5.000 μg/L at 24, 48 and 96 h, respectively; renal regeneration is evident (orange dotted circles) and there is a mass infiltration of hematopoietic cells around the newly formed tubules. White dotted circles in (**H**), shows glomeruli.

**Figure 2 ijms-26-04216-f002:**
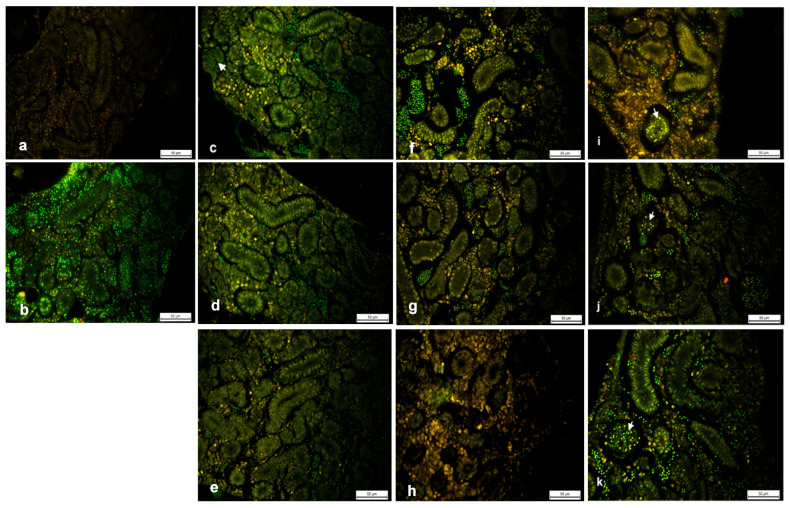
Acridine orange (AO) stained sections of zebrafish kidney after exposure to various concentrations of silver nanoparticles (AgNPs) for 24 to 96 h. (**a**), negative control lacking positive AO staining; (**b**), control section with very few necrotic nuclei; (**c**–**e**), cell nuclei stained positively with AO; scattered yellow fluorescence at 0.031 μg/L of AgNP exposure after 24, 48 and 96 h respectively; (**f**–**h**), cell nuclei stained positively with AO; aggregates of orange, fluorescent nuclei were detected at 0.250 μg/L of AgNP exposure and at 5.000 μg/L after 24 h (**i**); (**j**), orange fluorescence is minimal at 5.000 μg/L after 48 h; (**k**), cell nuclei stain positive with AO; green fluorescence is detected at 96 h of exposure. White arrows indicate the glomeruli. (Scale = bar is 50 μm).

## Data Availability

Additional data supporting the manuscript are available from the corresponding author upon request.

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
