# Peer review of "Silver Nanoparticle-Induced Nephrotoxicity in Zebrafish (Danio rerio)"

_ijms, 2025, doi:10.3390/ijms26094216_

Round 1
Reviewer 1 Report
Comments and Suggestions for Authors
The authors have submitted a detailed article, which mainly focus on the nephrotoxicity of silver nanoparticles in zebrafish. Even the Ag NPs were ‘green’ synthesized, significant toxicity were observed in the kidney of zebrafish. The topic is meaningful, but some issues should be carefully concerned.
- Section 2.2, the Ag NPs were observed by TEM and HRTEM, please provide the images in the main text. The formation of AgNPs was investigated using UV-vis spectroscopy, the UV results should also be provided.
- Three concentrations of Ag NPs were studied in this study, what is the basis for choosing these concentrations? Ag NPs is widely used as antibacterial agents, may be such concentration is not enough to inhibit the growth of bacteria. Please add the basis for concentration selection to the main text.
- Fig. 2, some numbers are missing, and the scale bar is not clear.
- The unit, ‘μg/ L-1’, should be ‘μg/L’.
Author Response
A point-by-point Response to Comments and Suggestions for Authors
Reviewer 1 Comments: [Comment: Section 2.2, the Ag NPs were observed by TEM and HRTEM, please provide the images in the main text. The formation of AgNPs was investigated using UV-vis spectroscopy, the UV results should also be provided]
Response 1: [Here, our PI of the project, who taught us how to synthesize and characterize the nanoparticles, had published the paper on the synthesis and characterization, so we cannot include the images in our manuscript but reference the paper to avoid plagiarism.
“Oluwafemi, O.S., et al., Green controlled synthesis of monodispersed, stable and smaller sized starch-capped silver nanoparticles. Materials Letters, 2013. 106: p. 332-336”.
Perhaps you can assist us in rephrasing our methods section of the manuscript.
Thank you for pointing this out. We agree with this comment, but we cannot publish the images as this was not the primary objective of the current study. It was done to us by the PI of the published paper shown above. Therefore, we need your input on this. However, we have added a statement in the relevant section of the manuscript to solve the issue.
Comments 2: [Reviewer 1 Comment: [Three concentrations of Ag NPs were studied in this study; what is the basis for choosing these concentrations].
Response 2: Agree. We used 3 concentrations.
Here, the Control (0 μg/L): This is essential to establish baseline biological responses without AgNP exposure. It allows for comparison with the treated groups to determine if any effects are indeed due to the nanoparticles.
Low (0.031 μg/L): This concentration aims to represent a potentially low level of environmental exposure or a concentration slightly above the detection limits in some environmental matrices.
Medium (0.250 μg/L): This concentration is an intermediate level to explore the dose-response relationship.
High (5.000 μg/L): This concentration is intended to elicit more pronounced effects, potentially approaching acute toxicity levels for some sensitive organisms.
Generally, the concentrations were based on the previously published papers on the LC50 value of AgNPs for 96 h in Japanese medaka (Oryzias latipes) was 0.9 mg/ L, Chae et al, 2009), the value for adult and larvae of zebrafish @7.07 mg/ L and 7.20 mg/L respectively (Griffitt et al ., 2008), and for fathead minnow (Pimephales promelas) @ 1.36 mg/ L. See Page 3 Section 2.3 of the manuscript.
Reviewer 1 Comment: In fig. 2, some numbers are missing, and the scale bar is unclear.
Response: These have been modified and enhanced to clarify the figure numbers. (See manuscript; the changes are highlighted in blue).
Reviewer 1 Comment: The unit, μg/ L-1’, should be ‘μg/L’.
Response: These corrections have been implemented. (See manuscript; the changes are highlighted in blue).

Reviewer 2 Report
Comments and Suggestions for Authors
Does the introduction provide sufficient background and include all relevant references?
it must required last 5 years of references more appreciated
Toxicity testing of AgNPs: AgNP solutions AgNP explain the size of Nanoparticles suitable and more explanation on oral intake ?
Toxicity explanation on ph?
Author Response
Reviewer 2 Comment: Does the introduction provide sufficient background and include all relevant references?
Response: Yes, it does. And some have been added.
Reviewer 2 Comment: it must require the last 5 years of references to be more appreciated.
Response: Some new references between the year2020-2025 have been added.
Reviewer 2 Comment: Toxicity testing of AgNPs: AgNP solutions AgNP explain the size of Nanoparticles suitable and more explanation on oral intake?
Response: This information has been added to the discussion section. This is highlighted in blue.
Reviewer 2 Comment: Toxicity explanation on ph?
Response: This information has been added to the discussion section
